# Clustering Stable Instances of Euclidean $k$-means

**Abhratanu Dutta**[*]
Northwestern University
adutta@u.northwestern.edu

**Aravindan Vijayaraghavan**[*]
Northwestern University
aravindv@northwestern.edu

**Alex Wang**[†]
Carnegie Mellon University
alexwang@u.northwestern.edu

## Abstract

The Euclidean $k$-means problem is arguably the most widely-studied clustering problem in machine learning. While the $k$-means objective is NP-hard in the worst-case, practitioners have enjoyed remarkable success in applying heuristics like Lloyd's algorithm for this problem. To address this disconnect, we study the following question: *what properties of real-world instances will enable us to design efficient algorithms and prove guarantees for finding the optimal clustering?* We consider a natural notion called additive perturbation stability that we believe captures many practical instances of Euclidean $k$-means clustering. Stable instances have unique optimal $k$-means solutions that does not change even when each point is perturbed a little (in Euclidean distance). This captures the property that $k$-means optimal solution should be tolerant to measurement errors and uncertainty in the points. We design efficient algorithms that provably recover the optimal clustering for instances that are additive perturbation stable. When the instance has some additional separation, we can design a simple, efficient algorithm with provable guarantees that is also robust to outliers. We also complement these results by studying the amount of stability in real datasets, and demonstrating that our algorithm performs well on these benchmark datasets.

## 1 Introduction

One of the major challenges in the theory of clustering is to bridge the large disconnect between our theoretical and practical understanding of the complexity of clustering. While theory tells us that most common clustering objectives like $k$-means or $k$-median clustering problems are intractable in the worst case, many heuristics like Lloyd's algorithm or k-means++ seem to be effective in practice. In fact, this has led to the "CDNM" thesis [11, 9]: "Clustering is difficult only when it does not matter".

We try to address the following natural questions in this paper: *Why are real-world instances of clustering easy? Can we identify properties of real-world instances that make them tractable?*

We focus on the Euclidean $k$-means clustering problem where we are given $n$ points $X = \{x_1, \ldots, x_n\} \subset \mathbb{R}^d$, and we need to find $k$ centers $\mu_1, \mu_2, \ldots, \mu_k \in \mathbb{R}^d$ minimizing the objective $\sum_{x \in X} \min_{i \in [k]} \|x - \mu_i\|^2$. The $k$-means clustering problem is the most well-studied objective for clustering points in Euclidean space [18, 3]. The problem is NP-hard in the worst-case [14] even for $k = 2$, and a constant factor hardness of approximation is known for larger $k$ [5].

---

[*]Supported by the National Science Foundation (NSF) under Grant No. CCF-1637585.
[†]Part of the work was done while the author was at Northwestern University.

One way to model real-world instances of clustering problems is through *instance stability*, which is an implicit structural assumption about the instance. Practically interesting instances of $k$-means clustering problem often have a clear optimal clustering solution (usually the ground-truth clustering) that is stable: i.e., it remains optimal even under small perturbations of the instance. As argued in [7], clustering objectives like $k$-means are often just a proxy for recovering a ground-truth clustering that is close to the optimal solution. Instances in practice always have measurement errors, and optimizing the $k$-means objective is meaningful only when the optimal solution is stable to these perturbations.

This notion of stability was formalized independently in a pair of influential works [11, 7]. The predominant strand of work on instance stability assumes that the optimal solution is resilient to multiplicative perturbations of the distances [11]. For any $\gamma \geq 1$, a metric clustering instance $(X, d)$ on point set $X \subset \mathbb{R}^d$ and metric $d : X \times X \rightarrow \mathbb{R}_+$ is said to be $\gamma$-factor stable iff the (unique) optimal clustering $C_1, \ldots, C_k$ of $X$ remains the optimal solution for any instance $(X, d')$ where any (subset) of the the distances are increased by up to a $\gamma$ factor i.e., $d(x, y) \leq d'(x, y) \leq \gamma d(x, y)$ for any $x, y \in X$. In a series of recent works [4, 8] culminating in [2], it was shown that 2-factor perturbation stable (i.e., $\gamma \geq 2$) instances of $k$-means can be solved in polynomial time.

Multiplicative perturbation stability represents an elegant, well-motivated formalism that captures robustness to measurement errors for clustering problems in general metric spaces ($\gamma = 1.1$ captures relative errors of 10% in the distances). However, multiplicative perturbation stability has the following drawbacks in the case of Euclidean clustering problems:

- Measurement errors in Euclidean instances are better captured using additive perturbations. Uncertainty of $\delta$ in the position of $x, y$ leads to an additive error of $\delta$ in $\|x - y\|_2$, irrespective of how large or small $\|x - y\|_2$ is.

- The amount of stability $\gamma$ needed to enable efficient algorithms (i.e., $\gamma \geq 2$) often imply strong structural conditions, that are unlikely to be satisfied by many real-world datasets. For instance, $\gamma$-factor perturbation stability implies that every point is a multiplicative factor of $\gamma$ closer to its own center (say $\mu_i$) than to any other cluster center $\mu_j$.

- Algorithms that are known to have provable guarantees under multiplicative perturbation stability are based on single-linkage or MST algorithms that are very non-robust by nature. In the presence of a few outliers or noise, any incorrect decision in the lower layers gets propagated up to the higher levels.

In this work, we consider a natural additive notion of stability for Euclidean instances, when the optimal $k$-means clustering solution does not change even where each point is moved by a small Euclidean distance of up to $\delta$. Moving each point by at most $\delta$ corresponds to a small additive perturbation to the pairwise distances between the points[3]. Unlike multiplicative notions of perturbation stability [11, 4], this notion of additive perturbation is not scale invariant. Hence the normalization or scale of the perturbation is important.

Ackerman and Ben-David [1] initiated the study of additive perturbation stability when the distance between any pair of points can be changed by at most $\delta = \varepsilon \operatorname{diam}(X)$ with $\operatorname{diam}(X)$ being the *diameter* of the whole dataset. The algorithms take time $n^{O(k/\varepsilon^2)} = n^{O(k \operatorname{diam}^2(X)/\delta^2)}$ and correspond to polynomial time algorithms when $k, 1/\varepsilon$ are constants. However, this dependence of $k \operatorname{diam}^2(X)/\delta^2$ in the exponent is not desirable since the diameter is a very non-robust quantity — the presence of one outlier (that is even far away from the decision boundary) can increase the diameter arbitrarily. Hence, these guarantees are useful mainly when the whole instance lies within a small ball and for a small number of clusters [1, 10]. Our notion of additive perturbation stability will use a different scale parameter that is closely related to the distance between the centers, instead of the diameter $\operatorname{diam}(X)$. As we will see soon, our results for additive perturbation stability have no explicit dependence on the diameter, and allows instances to have potentially unbounded clusters (as in the case of far-way outliers). Further with some additional assumptions, we also obtain polynomial time algorithmic guarantees for large $k$.

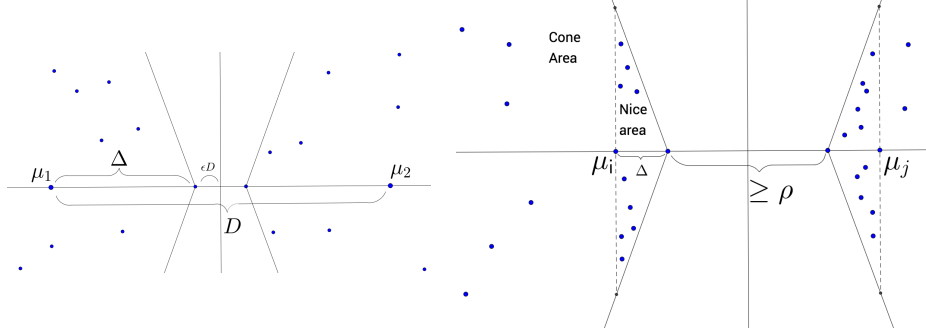

Figure 1: a)**Left:** the figure shows an instance with $k = 2$ satisfying $\varepsilon$-APS with $D$ being separation between the means. The half-angle of the cone is $\arctan(1/\varepsilon)$ and the distance between $\mu_1$ and the apex of the cone ($\Delta$) is at most $D/2$. b) **Right:** The figure shows a $(\rho, \Delta, \varepsilon)$-separated instance, with scale parameter $\Delta$. All the points lie inside the cones of half-angle $\arctan(1/\varepsilon)$, whose apexes are separated by a margin of at least $\rho$.

## 1.1  Additive Perturbation Stability and Our Contributions

We consider a notion of additive stability where the points in the instance can be moved by at most $\delta = \varepsilon D$, where $\varepsilon \in (0, 1)$ is a parameter, and $D = \max_{i \neq j} D_{ij} = \max_{i \neq j} \|\mu_i - \mu_j\|_2$ is the maximum distance between pairs of means. Suppose $X$ is a $k$-means clustering instance with optimal clustering $C_1, C_2, \ldots, C_k$. We say that $X$ is $\varepsilon$-APS (additive perturbation stable) iff every $\delta = \varepsilon D$-additive perturbation of $X$ has $C_1, C_2, \ldots, C_k$ as an optimal clustering solution. (See Definition 2.3 for a formal definition). Note that there is no restriction on the diameter of the instance, or even the diameters of the individual clusters. Hence, our notion of additive perturbation stability allows the instance to be unbounded.

**Geometric property of $\varepsilon$-APS instances.** Clusters in the optimal solution of an $\varepsilon$-APS instance satisfy a natural geometric condition, that implies an **"angular separation"** between every pair of clusters.

**Proposition 1.1** (Geometric Implication of $\varepsilon$-APS). *Consider an $\varepsilon$-APS instance $X$, and let $C_i, C_j$ be two clusters of the optimal solution. Any point $x \in C_i$ lies in a cone whose axis is along the direction $(\mu_i - \mu_j)$ with half-angle $\arctan(1/\varepsilon)$. Hence if $u$ is the unit vector along $\mu_i - \mu_j$ then*

$$\forall x \in C_i, \ \frac{|\langle x - \frac{\mu_i + \mu_j}{2}, u \rangle|}{\|x - \frac{\mu_i + \mu_j}{2}\|_2} > \frac{\varepsilon}{\sqrt{1 + \varepsilon^2}}. \tag{1}$$

For any $j \in [k]$, all the points in cluster $C_i$ lie inside the cone with its axis along $(\mu_i - \mu_j)$ as in Figure 1. The distance between $\mu_i$ and the apex of the cone is $\Delta = (\frac{1}{2} - \varepsilon)D$. We will call $\Delta$ the **scale parameter** of the clustering.

We believe that many clustering instances in practice satisfy $\varepsilon$-APS condition for reasonable constants $\varepsilon$. In fact, our experiments in Section 4 suggest that the above geometric condition is satisfied for reasonable values e.g., $\varepsilon \in (0.001, 0.2)$.

While the points can be arbitrarily far away from their own means, the above angular separation (1) is crucial in proving the polynomial time guarantees for our algorithms. For instance, this implies that at least $1/2$ of the points in a cluster $C_i$ are within a Euclidean distance of at most $O(\Delta/\varepsilon)$ from $\mu_i$. This geometric condition (1) of the dataset enables the design of a tractable algorithm for $k = 2$ with provable guarantees. This algorithm is based on a modification of the perceptron algorithm in supervised learning, and is inspired by [13].

**Informal Theorem 1.2.** *For any fixed $\varepsilon > 0$, there exists an $dn^{\mathrm{poly}(1/\varepsilon)}$ time algorithm that correctly clusters all $\varepsilon$-APS 2-means instances.*

For $k$-means clustering, similar techniques can be used to learn the separating halfspace for each pair of clusters. But this incurs an exponential dependence on $k^2$, which renders this approach

inefficient for large $k$.[4] We now consider a natural strengthening of this assumption that allows us to get $\text{poly}(n, d, k)$ guarantees.

**Angular Separation with additional Margin Separation.** We consider a natural strengthening of additive perturbation stability where there is an additional margin between any pair of clusters. This is reminiscent of margin assumptions in supervised learning of halfspaces and spectral clustering guarantees of Kumar and Kannan [15] (see Section 1.2). Consider a $k$-means clustering instance $X$ having an optimal solution $C_1, C_2, \ldots, C_k$. This instance is $(\rho, \Delta, \varepsilon)$-separated iff for each $i \neq j \in [k]$, the subinstance induced by $C_i, C_j$ has parameter scale $\Delta$, and all points in the clusters $C_i, C_j$ lie inside cones of half-angle $\arctan(1/\varepsilon)$, which are separated by a margin of at least $\rho$. This is implied by the stronger condition that the subinstance induced by $C_i, C_j$ is $\varepsilon$-additive perturbation stable with scale parameter $\Delta$ even when $C_i$ and $C_j$ are moved towards each other by $\rho$. Please see Figure 1 for an illustration. We formally define $(\rho, \Delta, \varepsilon)$-separated stable instances geometrically in Section 2.

**Informal Theorem 1.3** (Polytime algorithm for $(\rho, \Delta, \varepsilon)$-separated instances). *There is an algorithm running in time[5] $\widetilde{O}(n^2 kd)$ that given any instance $X$ that is $(\rho, \Delta, \varepsilon)$-separated with $\rho \geq \Omega(\Delta/\varepsilon^2)$ recovers the optimal clustering $C_1, \ldots, C_k$.*

A formal statement of the theorem (with unequal sized clusters), and its proof are given in Section 3. We prove these polynomial time guarantees for a new, simple algorithm ( Algorithm 3.1 ). The algorithm constructs a graph with one vertex for each point, and edges between points that within a distance of at most $r$ (for an appropriate threshold $r$). The algorithm then finds the $k$-largest connected components. It then uses the $k$ empirical means of these $k$ components to cluster all the points.

In addition to having provable guarantees, the algorithm also seems efficient in practice, and performs well on standard clustering datasets. Experiments that we conducted on some standard clustering datasets in UCI suggest that our algorithm manages to almost recover the ground truth, and achieves a $k$-means objective cost that is very comparable to Lloyd's algorithm and $k$-means++ (see Section 4).

In fact, our algorithm can also be used to initialize the Lloyd's algorithm: our guarantees show that when the instance is $(\rho, \Delta, \varepsilon)$-separated, one iteration of Lloyd's algorithm already finds the optimal clustering. Experiments suggest that our algorithm finds initializers of smaller $k$-means cost compared to the initializers of $k$-means++ [3] and also recover the ground-truth to good accuracy (see Section 4 and Supplementary material for details).

**Robustness to Outliers.** Perturbation stability requires the optimal solution to remain completely unchanged under any valid perturbation. In practice, the stability of an instance may be dramatically reduced by a few outliers. We show provable guarantees for a slight modification of the algorithm, in the setting when an $\eta$-fraction of the points can be arbitrary outliers, and do not lie in the stable regions. More formally, we assume that we are given an instance $X \cup Z$ where there is an (unknown) set of points $Z$ with $|Z| \leq \eta|X|$ such that $X$ is a $(\rho, \Delta, \varepsilon)$-separated-stable instance. Here $\eta n$ is assumed to be smaller than size of the smallest cluster by a constant factor. This is similar to robust perturbation resilience considered in [8, 16]. Our experiments in Section 4 indicate that the stability or separation can increase a lot after ignoring a few points close to the margin.

In what follows, $w_{\max} = \max|C_i|/n$ and $w_{\min} = \min|C_i|/n$ are the maximum and minimum weight of clusters, and $\eta < w_{\min}$.

**Theorem 1.4.** *Given $X \cup Z$ where $X$ satisfies $(\rho, \Delta, \varepsilon)$-separated with $\eta < w_{min}$ and*

$$\rho = \Omega\left(\frac{\Delta}{\varepsilon^2}\left(\frac{w_{\max} + \eta}{w_{\min} - \eta}\right)\right)$$

*and $\eta < w_{\min}$, there is a polynomial time algorithm running in time $\widetilde{O}(n^2 dk)$ that returns a clustering consistent with $C_1, \ldots, C_k$ on $X$.*

This robust algorithm is effectively the same as Algorithm 3.1 with one additional step that removes all low-degree vertices in the graph. This step removes bad outliers in $Z$ without removing too many points from $X$.

## 1.2 Comparisons to Other Related Work

Awasthi et al. [4] showed that $\gamma$-multiplicative perturbation stable instance also satisfied the notion of $\gamma$-center based stability (every point is a $\gamma$-factor closer to its center than to any other center) [4]. They showed that an algorithm based on the classic single linkage algorithm works under this weaker notion when $\gamma \geq 3$. This was subsequently improved by [8], and the best result along these lines [2] gives a polynomial time algorithm that works for $\gamma \geq 2$. A robust version of $(\gamma, \eta)$-perturbation resilience was explored for center-based clustering objectives [8]. As such, the notions of additive perturbation stability, and $(\rho, \Delta, \varepsilon)$-separated instances are incomparable to the various notions of multiplicative perturbation stability. Further as argued in [9], we believe that additive perturbation stability is more realistic for Euclidean clustering problems.

Ackerman and Ben-David [1] initiated the study of various deterministic assumptions for clustering instances. The measure of stability most related to this work is Center Perturbation (CP) clusterability (an instance is $\delta$-CP-clusterable if perturbing the centers by a distance of $\delta$ does not increase the cost much). A subtle difference is their focus on obtaining solutions with small objective cost [1], while our goal is to recover the optimal clustering. However, the main qualitative difference is how the length scale is defined – this is crucial for additive perturbations. The run time of the algorithm in [1] is $n^{poly(k,\mathrm{diam}(X)/\delta)}$ , where the length scale of the perturbations is $\mathrm{diam}(X)$, the diameter of the whole instance. Our notion of additive perturbations uses a much smaller length-scale of $\Delta$ (essentially the inter-mean distance; see Prop. 1.1 for a geometric intepretation), and Theorem 1.2 gives a run-time guarantee of $n^{poly(\Delta/\delta)}$ for $k = 2$ (Theorem 1.2 is stated in terms of $\varepsilon = \delta/\Delta$). By using the largest inter-mean distance instead of the diameter as the length scale, our algorithmic guarantees can also handle unbounded clusters with arbitrarily large diameters and outliers.

The exciting results of Kumar and Kannan [15] and Awasthi and Sheffet [6] also gave deterministic margin-separation condition, under which spectral clustering (PCA followed by $k$-means) finds the optimum clusters under deterministic conditions about the data. Suppose $\sigma = \|X - C\|_{op}^2/n$ is the "spectral radius" of the dataset, where $C$ is the matrix given by the centers. In the case of equal-sized clusters, the improved results of [6] proves approximate recovery of the optimal clustering if the margin $\rho$ between the clusters along the line joining the centers satisfies $\rho = \Omega(\sqrt{k}\sigma)$. Our notion of margin $\rho$ in $(\rho, \Delta, \varepsilon)$-separated instances is analogous to the margin separation notion used by the above results on spectral clustering [15, 6]. In particular, we require a margin of $\rho = \Omega(\Delta/\varepsilon^2)$ where $\Delta$ is our scale parameter, with no extra $\sqrt{k}$ factor. However, we emphasize that the two margin conditions are incomparable, since the spectral radius $\sigma$ is incomparable to the scale parameter $\Delta$.

We now illustrate the difference between these deterministic conditions by presenting a couple of examples. Consider an instance with $n$ points drawn from a mixture of $k$ Gaussians in $d$ dimensions with identical diagonal covariance matrices with variance 1 in the first $O(1)$ coordinates and roughly $\frac{1}{d}$ in the others, and all the means lying in the subspace spanned by these first $O(1)$ co-ordinates. In this setting, the results of [15, 6] require a margin separation of at least $\sqrt{k \log n}$ between clusters. On the other hand, these instances satisfy our geometric conditions with $\varepsilon = \Omega(1)$, $\Delta \sim \sqrt{\log n}$ and therefore our algorithm only needs a margin separation of $\rho\sqrt{\log n}$ ( hence, saving a factor of $\sqrt{k}$). [6] However, if the $n$ points were drawn from a mixture of spherical Gaussians in high dimensions (with $d \gg k$), then the margin condition required for [15, 6] is weaker.

## 2 Stability definitions and geometric properties

$X \subseteq \mathbb{R}^d$ will denote a $k$-means clustering instance and $C_1, \ldots, C_k$ will often refer to its optimal clustering. It is well-known that given a cluster $C$ the value of $\mu$ minimizing $\sum_{x \in C} \|x - \mu\|^2$ is given by $\mu = \frac{1}{|C|} \sum_{x \in C} x$, the mean of the points in the set. We give more preliminaries about the $k$-means problem in the Supplementary Material.

### 2.1 Balance Parameter

We define an instance parameter, $\beta$, capturing how balanced a given instance's clusters are.

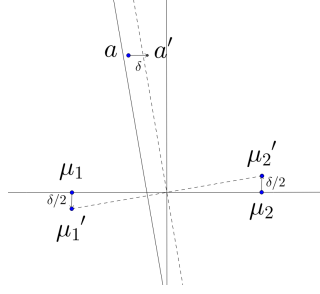

Figure 2: An example of the family of perturbations considered by Lemma 2.4. Here $v$ is in the upwards direction. If $a$ is to the right of the diagonal solid line, then $a'$ will be to the right of the slanted dashed line and will lie on the wrong side of the separating hyperplane.

**Definition 2.1** (Balance parameter). Given an instance $X$ with optimal clustering $C_1, \ldots, C_k$, we say $X$ satisfies balance parameter $\beta \geq 1$ if for all $i \neq j$, $\beta |C_i| > |C_j|$.

We will write $\beta$ in place of $\beta(X)$ when the instance is clear from context.

## 2.2 Additive perturbation stability

**Definition 2.2** ($\varepsilon$-additive perturbation). Let $X = \{x_1, \ldots, x_n\}$ be a $k$-means clustering instance with optimal clustering $C_1, C_2, \ldots, C_k$ whose means are given by $\mu_1, \mu_2, \ldots, \mu_k$. Let $D = \max_{i,j} \|\mu_i - \mu_j\|$. We say that the instance $X' = \{x'_1, \ldots, x'_n\}$ is an $\varepsilon$-additive perturbation of $X$ if for all $i$, $\|x'_i - x_i\| \leq \varepsilon D$.

**Definition 2.3** ($\varepsilon$-additive perturbation stability). Let $X$ be a $k$-means clustering instance with optimal clustering $C_1, C_2, \ldots, C_k$. We say that $X$ is $\varepsilon$-additive perturbation stable (APS) if every $\varepsilon$-additive perturbation of $X$ has unique optimal clustering given by $C_1, C_2, \ldots, C_k$.

Intuitively, the difficulty of the clustering task increases as the stability parameter $\varepsilon$ decreases. For example, when $\varepsilon = 0$ the set of $\varepsilon$-APS instances contains any instance with a unique solution. In the following we will only consider $\varepsilon > 0$.

## 2.3 Geometric implication of $\varepsilon$-APS

Let $X$ be an $\varepsilon$-APS $k$-means clustering instance such that each cluster has at least 4 points. Fix $i \neq j$ and consider a pair of clusters $C_i, C_j$ with means $\mu_i, \mu_j$ and define the following notations.

- Let $D_{i,j} = \|\mu_i - \mu_j\|$ be the distance between $\mu_i$ and $\mu_j$ and let $D = \max_{i',j'} \|\mu_{i'} - \mu_{j'}\|$ be the maximum distance between any pair of means.

- Let $u = \frac{\mu_i - \mu_j}{\|\mu_i - \mu_j\|}$ be the unit vector in the intermean direction. Let $V = u^\perp$ be the space orthogonal to $u$. For $x \in \mathbb{R}^d$, let $x_{(u)}$ and $x_{(V)}$ be the projections $x$ onto $u$ and $V$.

- Let $p = \frac{\mu_i + \mu_j}{2}$ be the midpoint between $\mu_i$ and $\mu_j$.

A simple perturbation that we can use will move all points in $C_i$ and $C_j$ along the direction $\mu_i - \mu_j$ by a $\delta$ amount, while another perturbation moves these points along $\mu_j - \mu_i$; these will allow us to conclude that $\exists$ margin of size $2\delta$. To establish Proposition 1.1, we will choose a clever $\varepsilon$-perturbation that allows us to show that clusters must live in *cone* regions (see figure 1 left). This perturbation chooses two clusters and moves their means in opposite directions orthogonal to $u$ while moving a single point towards the other cluster (see figure 2). The following lemma establishes Proposition 1.1.

**Lemma 2.4.** *For any $x \in C_i \cup C_j$, $\|(x - p)_{(V)}\| < \frac{1}{\varepsilon} \left( \|(x - p)_{(u)}\| - \varepsilon D_{i,j} \right)$.*

*Proof.* Let $v \in V$ be a unit vector perpendicular to $u$. Without loss of generality, let $a, b, c, d \in C_i$ be distinct. Note that $D_{i,j} \leq D$ and consider the $\varepsilon$-additive perturbation given by

$$X' = \{a - \delta u, b + \delta u, c - \delta v, d - \delta v\} \cup \{x - \tfrac{\delta}{2}v \mid x \in C_i \setminus \{a, b, c, d\}\} \cup \{x + \tfrac{\delta}{2}v \mid x \in C_j\}$$

and $X \setminus \{C_i \cup C_j\}$ where $\delta = \varepsilon D_{i,j}$ (see figure 2). By assumption, $\{C_i, C_j\}$ remains the optimal clustering of $C_i \cup C_j$. We have constructed $X'$ such that the new means are at $\mu_i' = \mu_i - \frac{\varepsilon D_{i,j}}{2} v$ and $\mu_j' = \mu_j + \frac{\varepsilon D_{i,j}}{2} v$, and the midpoint between the means is $p' = p$. The halfspace containing $\mu_i'$ given by the linear separator between $\mu_i'$ and $\mu_j'$ is $\langle x - p', \mu_i' - \mu_j' \rangle > 0$. Hence, as $a'$ is classified correctly by the $\varepsilon$-APS assumption,

$$\langle a' - p', \mu_i' - \mu_j' \rangle = \langle a - p - \varepsilon D_{i,j} u, D_{i,j} u - \varepsilon D_{i,j} v \rangle$$
$$= D_{i,j} (\langle a - p, u \rangle - \varepsilon \langle a - p, v \rangle - \varepsilon D_{i,j}) > 0$$

Then noting that $\langle u, a - p \rangle$ is positive, we have that $\langle a - p, v \rangle < \frac{1}{\varepsilon} \left( \| (a-p)_{(u)} \| - \varepsilon D_{i,j} \right)$. $\quad\square$

Note that this property follows from perturbations which only affect two clusters at a time. Our results follow from this weaker notion.

### 2.4 $(\rho, \Delta, \varepsilon)$-separation

Motivated by Lemma 2.4, we define a geometric condition where the angular separation and margin separation are parametrized separately. This notion of separation is implied by a stronger stability assumption where any pair of clusters is $\varepsilon$-APS with scale parameter $\Delta$ even after being moved towards each other a distance of $\rho$.

We say that a pair of clusters is $(\rho, \Delta, \varepsilon)$-separated if their points lie in cones with axes along the intermean direction, half-angle $\arctan(1/\varepsilon)$, and apexes at distance $\Delta$ from their means and at least $\rho$ from each other (see figure 1 right). Formally, we require the following.

**Definition 2.5** (Pairwise $(\rho, \Delta, \varepsilon)$-separation). Given a pair of clusters $C_i, C_j$ with means $\mu_i, \mu_j$, let $u = \frac{\mu_i - \mu_j}{\|\mu_i - \mu_j\|}$ be the unit vector in the intermean direction and let $p = (\mu_i + \mu_j)/2$. We say that $C_i$ and $C_j$ are $(\rho, \Delta, \varepsilon)$-separated if $D_{i,j} \geq \rho + 2\Delta$ and for all $x \in C_i \cup C_j$,

$$\| (x - p)_{(V)} \| \leq \frac{1}{\varepsilon} \left( \| (x - p)_{(u)} \| - (D_{i,j}/2 - \Delta) \right).$$

**Definition 2.6** ($(\rho, \Delta, \varepsilon)$-separation). We say that an instance $X$ is $(\rho, \Delta, \varepsilon)$-separated if every pair of clusters in the optimal clustering is $(\rho, \Delta, \varepsilon)$-separated.

## 3 $k$-means clustering for general $k$

We assume that our instance has balance parameter $\beta$. Our algorithm takes in as input the set of points $X$ and $k$, and outputs a clustering of all the points.

**Algorithm 3.1.**

**Input:** $X = \{x_1, \ldots, x_n\}$, $k$.
 1: **for all** pairs $a, b$ of distinct points in $\{x_i\}$ **do**
 2: $\quad$ Let $r = \|a - b\|$ be our guess for $\rho$
 3: $\quad$ **procedure** INITIALIZE
 4: $\quad\quad$ Create graph $G$ on vertex set $\{x_1, \ldots, x_n\}$ where $x_i$ and $x_j$ have an edge iff $\|x_i - x_j\| < r$
 5: $\quad\quad$ Let $a_1, \ldots, a_k \in \mathbb{R}^d$ where $a_i$ is the mean of the $i$th largest connected component of $G$
 6: $\quad$ **procedure** ASSIGN
 7: $\quad\quad$ Let $C_1, \ldots, C_k$ be the clusters obtained by assigning each point in $X$ to the closest $a_i$
 8: $\quad$ Calculate the $k$-means objective of $C_1, \ldots, C_k$
 9: Return clustering with smallest $k$-means objective found above

**Theorem 3.2.** *Algorithm 3.1 recovers $C_1, \ldots, C_k$ for any $(\rho, \Delta, \varepsilon)$-separated instance with $\rho = \Omega \left( \frac{\Delta}{\varepsilon^2} + \frac{\beta \Delta}{\varepsilon} \right)$ and the running time is $\widetilde{O}(n^2 kd)$.*

We maintain the connected components and their centers via a union-find data structure and keep it updated as we increase $\rho$ and add edges to the dynamic graph. Since we go over $n^2$ possible choices of $\rho$ and each pass takes $O(kd)$ time, the algorithm runs in $\widetilde{O}(n^2 kd)$.

The rest of the section is devoted to proving Theorem 3.2. Define the following regions of $\mathbb{R}^d$ for every pair $i, j$. Given $i, j$, let $C_i, C_j$ be the corresponding clusters with means $\mu_i, \mu_j$. Let $u = \frac{\mu_i - \mu_j}{\|\mu_i - \mu_j\|}$ be the unit vector in the inter-mean direction and $p = \frac{\mu_i + \mu_j}{2}$ be the point between the two means. We first define formally $S_{i,j}^{(\text{cone})}$ which was described in the introduction (the feasible region) and two other regions of the clusters that will be useful in the analysis (see Figure 1b). We observe that $C_i$ belongs to the intersection of all the cones $S_{i,j}^{(\text{cone})}$ over $j \neq i$.

**Definition 3.3.**

- $S_{i,j}^{(\text{cone})} = \{\, x \in \mathbb{R}^d \mid \|(x - (\mu_i - \Delta u))_{(V)}\| \leq \frac{1}{\varepsilon}\langle x - (\mu_i - \Delta u), u\rangle \,\}$,

- $S_{i,j}^{(\text{nice})} = \{\, x \in S_{i,j}^{(\text{cone})} \mid \langle x - \mu_i, u\rangle \leq 0 \,\}$,

- $S_i^{(\text{good})} = \bigcap_{j \neq i} S_{i,j}^{(\text{nice})}$.

The nice area of $i$ with respect to $j$ i.e. $S_{i,j}^{(\text{nice})}$, is defined as all points in the cap of $S_{i,j}^{(\text{cone})}$ "above" $\mu_i$. The good area of a cluster $S_i^{(\text{good})}$ is the intersection of its nice areas with respect to all other clusters.

It suffices to prove the following two main lemmas. Lemma 3.4 states that the `ASSIGN` subroutine correctly clusters all points given an initialization satisfying certain properties. Lemma 3.5 states that the initialization returned by the `INITIALIZE` subroutine satisfies these properties when we guess $r = \rho$ correctly. As $\rho$ is only used as a threshold on edge lengths, testing the distances between all pairs of data points i.e. $\{\,\|a - b\| : a, b \in X\,\}$ suffices.

**Lemma 3.4.** *For a $(\rho, \Delta, \varepsilon)$-separated instance with $\rho = \Omega(\Delta/\varepsilon^2)$, the `ASSIGN` subroutine recovers $C_1, C_2, \cdots C_k$ correctly when initialized with $k$ points $\{\, a_1, a_2, \ldots, a_k \,\}$ where $a_i \in S_i^{(\text{good})}$.*

**Lemma 3.5.** *For an $(\rho, \Delta, \varepsilon)$-separated instance with balance parameter $\beta$ and $\rho = \Omega(\beta\Delta/\varepsilon)$, the `INITIALIZE` subroutine outputs one point each from $\{\, S_i^{(good)} : i \in [k]\,\}$ when $r = \rho$.*

To prove Lemma 3.5 we define a region of each cluster $S_i^{(\text{core})}$, the core, such that most (at least $\beta/(1 + \beta)$ fraction) of the points in $C_i$ will belong to the connected component containing $S_i^{(\text{core})}$. Hence, any large connected component (in particular, the $k$ largest ones) must contain the core of one of the clusters. Meanwhile, the margin ensures points across clusters are not connected. Further, since $S_i^{(\text{core})}$ accounts for most points in $C_i$, the angular separation ensures that the empirical mean of the connected component is in $S_i^{(\text{good})}$.

# 4 Experimental results

We evaluate Algorithm 3.1 on multiple real world datasets and compare its performance to the performance of $k$-means++, and also check how well these datasets satisfy our geometric conditions. See supplementary results for details about ground truth recovery.

**Datasets.** Experiments were run on unnormalized and normalized versions of four labeled datasets from the UCI Machine Learning Repository: Wine ($n = 178, k = 3, d = 13$), Iris ($n = 150, k = 3, d = 4$), Banknote Authentication ($n = 1372, k = 2, d = 5$), and Letter Recognition ($n = 20,000, k = 26, d = 16$). Normalization was used to scale each feature to unit range.

**Performance** We ran Algorithm 3.1 on the datasets. The cost of the returned solution for each of the normalized and unnormalized versions of the datasets is recorded in Table 1 column 2. Our guarantees show that under $(\rho, \Delta, \varepsilon)$-separation for appropriate values of $\rho$ (see section 3), the algorithm will find the optimal clustering after a single iteration of Lloyd's algorithm. Even when $\rho$ does not satisfy our requirement, we can use our algorithm as an initialization heuristic for Lloyd's algorithm. We compare our initialization with the $k$-means++ initialization heuristic ($D^2$ weighting). In Table 1, this is compared to the smallest initialization cost of 1000 trials of $k$-means++ on each of the datasets, the solution found by Lloyd's algorithm using our initialization and the smallest $k$-means cost of 100 trials of Lloyd's algorithm using a $k$-mean++ initialization.

**Separation in real data sets.** As the ground truth clusterings in our datasets are not in general linearly separable, we consider the clusters given by Lloyd's algorithm initialized with the ground

Table 1: Comparison of $k$-means cost for Alg 3.1 and $k$-means++

| Dataset | Alg 3.1 | $k$-means++ | Alg 3.1 with Lloyd's | $k$-means++ with Lloyd's |
|---|---|---|---|---|
| Wine | 2.376e+06 | 2.426e+06 | 2.371e+06 | 2.371e+06 |
| Wine (normalized) | 48.99 | 65.50 | 48.99 | 48.95 |
| Iris | 81.04 | 86.45 | 78.95 | 78.94 |
| Iris (normalized) | 7.035 | 7.676 | 6.998 | 6.998 |
| Banknote Auth. | 44808.9 | 49959.9 | 44049.4 | 44049.4 |
| Banknote (norm.) | 138.4 | 155.7 | 138.1 | 138.1 |
| Letter Recognition | 744707 | 921643 | 629407 | 611268 |
| Letter Rec. (norm.) | 3367.8 | 4092.1 | 2767.5 | 2742.3 |

Table 2: Values of $(\rho, \varepsilon, \Delta)$ satisfied by $(1 - \eta)$-fraction of points

| Dataset | $\eta$ | $\varepsilon$ | minimum $\rho/\Delta$ | average $\rho/\Delta$ | maximum $\rho/\Delta$ |
|---|---|---|---|---|---|
| Wine | 0.1 | 0.1 | 0.566 | 1.5 | 3.05 |
| | | 0.01 | 0.609 | 1.53 | 3.07 |
| Iris | 0.1 | 0.1 | 0.398 | 4.35 | 7.7 |
| | | 0.01 | 0.496 | 5.04 | 9.06 |
| Banknote Auth. | 0.1 | 0.1 | 0.264 | 0.264 | 0.264 |
| | | 0.01 | 0.398 | 0.398 | 0.398 |
| Letter Recognition | 0.1 | 0.1 | 0.018 | 2.19 | 7.11 |
| | | 0.01 | 0.378 | 3.07 | 11.4 |

truth solutions. *Values of $\varepsilon$ for Lemma 2.4.* We calculate the maximum value of $\varepsilon$ such that a given pair of clusters satisfies the geometric condition in Proposition 1.1. The results are collected in the Supplementary material in Table 3. We see that the average value of $\varepsilon$ lies approximately in the range $(0.01, 0.1)$.

*Values of $(\rho, \Delta, \varepsilon)$-separation.* We attempt to measure the values of $\rho$, $\Delta$, and $\varepsilon$ in the datasets. For $\eta = 0.05, 0.1$, $\varepsilon = 0.1, 0.01$, and a pair of clusters $C_i$, $C_j$, we calculate $\rho$ as the maximum margin separation a pair of axis-aligned cones with half-angle $\arctan(1/\varepsilon)$ can have while capturing a $(1 - \eta)$-fraction of all points. For some datasets and values for $\eta$ and $\varepsilon$, there may not be any such value of $\rho$, in this case we leave the row blank. The results for the unnormalized datasets with $\eta = 0.1$ are collected in Table 2. (See Supplementary material for the full table).

## 5 Conclusion and Future Directions

We studied a natural notion of additive perturbation stability, that we believe captures many real-world instances of Euclidean $k$-means clustering. We first gave a polynomial time algorithm when $k = 2$. For large $k$, under an additional margin assumption, we gave a fast algorithm of independent interest that provably recovers the optimal clustering under these assumptions (in fact the algorithm is also robust to noise and outliers). An appealing aspect of this algorithm is that it is not tailored towards the model; our experiments indicate that this algorithm works well in practice even when the assumptions do not hold. Our results with the margin assumption hence gives an algorithm that (A) has provable guarantees (under reasonable assumptions) (B) is efficient and practical and (C) is robust to errors.

While the margin assumption seems like a realistic assumption qualitatively, we believe that the exact condition we assume is not optimal. An interesting open question is understanding whether such a margin is necessary for designing tractable algorithms for large $k$. We conjecture that for higher $k$, clustering remains hard even with $\varepsilon$ additive perturbation resilience (without any additional margin assumption). Improving the margin condition or proving lower bounds on the amount of additive stability required are interesting future directions.

## Footnotes

[3]Note that not all additive perturbations to the distances can be captured by an appropriate movement of the points in the cluster. Hence the notion we consider in our paper is a weaker assumption on the instance.

[4]We remark that the results of [1] also incur an exponential dependence on $k$.

[5]The $\widetilde{O}$ hides an inverse Ackerman fuction of $n$.

[6]Further, while algorithms for learning GMM models may work here, adding some outliers far from the decision boundary will cause many of these algorithms to fail, while our algorithm is robust to such outliers.

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
