[Supplementary Material]

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

# Appendix A  Introduction to the $k$-means clustering problem

In the $k$-means clustering problem, we are given $n$ points $\{x_1, \ldots, x_n\}$ in $\mathbb{R}^d$ and need to find $k$ centers $\mu_1, \mu_2, \ldots, \mu_k \in \mathbb{R}^d$ minimizing

$$\sum_x \min_{i \in [k]} \|x - \mu_i\|^2.$$

Choosing the centers $\mu_1, \mu_2, \ldots, \mu_k$ fixes the clustering $C_1, C_2, \ldots, C_k$ given by $C_i = \{x \mid \forall j, \|x - \mu_i\| \leq \|x - \mu_j\|\}$. We can rewrite the objective as

$$\sum_{i \in [k]} \sum_{x \in C_i} \|x - \mu_i\|^2.$$

On the other hand, given a cluster $C_i$ the value of $\mu$ minimizing $\sum_{x \in C_i} \|x - \mu\|^2$ is given by $\mu = \frac{1}{|C_i|} \sum_{x \in C_i} x$, the mean of the points in the set. Thus, we can reformulate the problem as finding a clustering $C_1, C_2, \ldots, C_k$ of $\{x_i\}$ minimizing

$$\sum_{i \in [k]} \sum_{y \in C_i} \left\| y - \left( \frac{1}{|C_i|} \sum_{x \in C_i} x \right) \right\|^2.$$

# Appendix B  Exact poly-time algorithm for $\varepsilon$-APS $2$-means clustering

The following algorithm is inspired by work in [] showing that the perceptron algorithm runs in poly-time with high probability in the smoothed analysis setting.

## B.1  Review of perceptron algorithm

Suppose $y_1, \ldots, y_n$ is a sequence of labeled $\{+1, -1\}$-samples consistent with a linear threshold function, i.e., there exists vector $w^*$ such that the labeling function $\ell(y_i)$ is consistent with $\mathrm{sgn}(\langle w^*, y_i \rangle)$. At time $t = 0$, the perceptron algorithm sets $w_0 = 0$. At each subsequent time step, the algorithm sees sample $y_t$, outputs $\mathrm{sgn}(\langle w_{t-1}, y_t \rangle)$ as its guess for $\ell(y_t)$, sees the true label $\ell(y_t)$, and updates $w_t$. On a correct guess, $w_t = w_{t-1}$, and on a mistake $w_t = w_{t-1} + \ell(y_t) y_t / \|y_t\|$.

The following well-known theorem [12, 17] bounds the number of total mistakes the perceptron algorithm can make in terms of its angular margin.

**Theorem B.1.** *The number of mistakes made by the perceptron algorithm is bounded above by $(1/\gamma)^2$ for*

$$\gamma = \min_{i \in [n]} \frac{|\langle w^*, y_i \rangle|}{\|w^*\| \|y_i\|}.$$

For a universe $U$ of elements and a function $f : U \to \mathbb{Z}_{\geq 0}$, we will denote by $(U, f)$ the multiset where $u \in U$ appears in the multiset $f(u)$-many times. The size of a multiset is $\sum_{u \in U} f(u)$.

**Lemma B.2.** *There exists a multiset $M = (\{y_1, \ldots, y_n\}, f)$ of size at most $(1/\gamma)^2$ such that $\sum_{y \in M} \ell(y) \frac{y}{\|y\|}$ correctly classifies all of $\{y_1, \ldots, y_n\}$.*

*Proof.* Let $r = (1/\gamma)^2 + 1$. Consider the performance of the perceptron algorithm on $r$ consecutive runs of the $y_1, \ldots, y_n$, i.e., let the input be

$$\overbrace{y_1, \ldots, y_n}^{1 \text{ run}}, y_1, \ldots, y_n, \ldots, \underbrace{y_1, \ldots, y_n}_{r \text{ runs}}.$$

If no mistakes are made on a given run, no mistakes will be made on any subsequent run. Thus if a mistake is made on a given run, at least one mistake must have been made on every previous run. Suppose the perceptron algorithm makes a mistake on the $r$th run, then the algorithm must have made at least $(1/\gamma)^2 + 1$ mistakes, a contradiction. Hence the direction of $w$ after $r$ runs correctly classifies all of $\{y_1, \ldots, y_n\}$. The value of $w$ is $\sum_{i \in [n]} f(y_i) \ell(y_i) \frac{y_i}{\|y_i\|}$ where $f(y_i)$ is the number of times $y_i$ was misclassified. □

## B.2 The algorithm

Fix the following notation for this section: let $I = \{x_1, \ldots, x_n\}$ for $x_i \in \mathbb{R}^d$ be an $\varepsilon$-APS 2-means clustering instance such that each cluster has at least 4 points. Let $D = \|\mu_1 - \mu_2\|$, $u = \frac{\mu_1 - \mu_2}{\|\mu_1 - \mu_2\|}$, $p = \frac{\mu_1 + \mu_2}{2}$.

**Algorithm B.3.**

---

**Input:** $X = \{x_1, \ldots, x_n\}, \varepsilon$
1: **for all** pairs $a, b$ of distinct points in $\{x_i\}$ **do**
2:     Let $\delta = \|a - b\|$
3:     Let $X_{a,b} = \{y_1, \ldots, y_n\}$ be an instance given by $y_i = (x_i - a, \delta) \in \mathbb{R}^{d+1}$
4:     **for all** multisets $M$ of size at most $c_1^{-2}\varepsilon^{-8}$ and assignments $\ell : M \to \{\pm 1\}$ **do**
5:         Let $w = \sum_{y \in M} \ell(y)\frac{y}{\|y\|}$
6:         Calculate $k$-means cost of $C_1 = \{x_i \mid \langle w, y_i \rangle \geq 0\}, C_2 = \{x_i \mid \langle w, y_i \rangle < 0\}$.
7: Return clustering with smallest $k$-means objective found above

---

## B.3 Geometric conditions

The following two lemmas follow directly from Lemma 2.4.

**Lemma B.4.** *For any $x \in C_i \cup C_j$,*

$$|\langle u, x - p \rangle| > \varepsilon D_{i,j}.$$

*In particular, for $x \in C_i$, $\langle u, x - p \rangle > \varepsilon D_{i,j}$ and for $x \in C_j$, $\langle u, x - p \rangle < -\varepsilon D_{i,j}$.*

**Lemma B.5.** *For any $x \in C_i \cup C_j$,*

$$\frac{|\langle u, x - p \rangle|}{\|x - p\|} > \sqrt{\frac{\varepsilon^2}{1 + \varepsilon^2}}.$$

The following statement says that the nice area of any cluster $C_i$ with respect to another cluster $C_j$ is nonempty.

**Lemma B.6.** *For any $i \neq j$, $S_{i,j}^{(nice)}$ is nonempty.*

*Proof.* We have that $\frac{1}{|C_i|} \sum_{x \in C_i}(x - p) = \mu_i - p = (\Delta_{i,j}/2)u$. Then

$$\frac{1}{|C_i|} \sum_{x \in C_i} \langle u, x - p \rangle = \Delta_{i,j}/2$$

where each term in the summation is positive. Thus there is some $x \in C_i$ such that $|\langle v, x - p \rangle| \leq \Delta_{i,j}/2$. □

Note that Lemmas B.4 and B.6 together imply that we *can not* have an instance with $\varepsilon \geq 1/2$.

**Lemma B.7.** *There is no $\varepsilon$-APS $k$-means clustering instance for $\varepsilon \geq 1/2$.*

## B.4 Proof of correctness of Algorithm B.3

Note that each new instance $X_{a,b}$ has labeling consistent with some linear threshold function. We have that $\ell(y_i) = \ell(x_i) = \text{sgn}(\langle u, x_i - p \rangle) = \text{sgn}(\langle u, x_i - a \rangle + \langle u, a - p \rangle)$. Then taking $w^* = (u, \langle u, a - p \rangle / \delta)$, we have that $\ell(y_i) = \text{sgn}(\langle w^*, y_i \rangle)$.

**Lemma B.8.** *Let $a \in S_{1,2}^{(nice)}, b \in S_{2,1}^{(nice)}$. Then,*

$$(2\varepsilon)D < \|a - b\| < \left(\sqrt{\frac{1 + \varepsilon^2}{\varepsilon^2}}\right)D.$$

*Proof.* For the first inequality, $\|a - b\| \geq |\langle u, a - b \rangle| = |\langle u, a - p \rangle - \langle u, b - p \rangle|$. Then by Lemma B.4, $\|a - b\| > 2\varepsilon D$.

For the second inequality, $\|a - b\| \leq \|a - p\| + \|p - b\|$. As $a \in S_{1,2}^{\text{(nice)}}$, we have $|\langle u, a - p \rangle| \leq \Delta/2$, hence by Lemma B.5, $\|a - p\| < \sqrt{(1 + \varepsilon^2)/\varepsilon^2} D/2$. Similarly, $\|b - p\| < \sqrt{(1 + \varepsilon^2)/\varepsilon^2} D/2$. $\square$

The following lemma states that the angular separation $\gamma$ in $I_{a,b}$ is large when $a \in S_{1,2}^{\text{(nice)}}$, $b \in S_{2,1}^{\text{(nice)}}$.

**Lemma B.9.** *There exists constant $c_1$ such that for any instance $I_{a,b} = \{y_i\}$ corresponding to $a \in S_{1,2}^{(nice)}$, $b \in S_{2,1}^{(nice)}$,*

$$\gamma = \min_{i \in [n]} \frac{|\langle w^*, y_i \rangle|}{\|w^*\| \, \|y_i\|} > c_1 \varepsilon^4.$$

*Proof.* We bound each term in the minimization individually. Let $i \in [n]$, then

$$\frac{|\langle w^*, y_i \rangle|}{\|w^*\| \, \|y_i\|} = \frac{|\langle u, x_i - p \rangle|}{\sqrt{1 + \left( \frac{\langle u, a - p \rangle}{\delta} \right)^2} \sqrt{\delta^2 + \|x_i - a\|^2}}.$$

We first observe the following facts.

- From Lemma B.5, $|\langle u, x_i - p \rangle| > \sqrt{\frac{\varepsilon^2}{1 + \varepsilon^2}} \|x_i - p\| > \varepsilon \|x_i - p\|$

- As $a$ is in $S_{1,2}^{\text{(nice)}}$, $|\langle u, a - p \rangle| \leq \frac{D}{2} < \frac{\delta}{4\varepsilon}$

- As $a$ is in $S_{1,2}^{\text{(nice)}}$, by Lemma B.5, $\|x_i - a\|^2 \leq 2 \|x_i - p\|^2 + 2 \|p - a\|^2 < 2 \|x_i - p\|^2 + \frac{1}{2} \frac{1 + \varepsilon^2}{\varepsilon^2} D^2$

- From Lemma B.8, $\delta^2 < \frac{1 + \varepsilon^2}{\varepsilon^2} D^2$,

- From Lemma B.4, $\|x_i - p\| > \varepsilon D$

Making each of the substitutions above,

$$\frac{|\langle w^*, y_i \rangle|}{\|w^*\| \, \|y_i\|} > \varepsilon \frac{\|x_i - p\|}{\sqrt{1 + \frac{1}{16\varepsilon^2}} \sqrt{2 \|x_i - p\|^2 + \frac{3}{2} \frac{1 + \varepsilon^2}{\varepsilon^2} D^2}}$$

$$> \varepsilon \frac{1}{\sqrt{1 + \frac{1}{16\varepsilon^2}} \sqrt{2 + \frac{3}{2} \frac{1 + \varepsilon^2}{\varepsilon^2} \left( \frac{D}{\|x_i - p\|} \right)^2}}$$

$$> \varepsilon \frac{1}{\sqrt{1 + \frac{1}{16\varepsilon^2}} \sqrt{2 + \frac{3}{2\varepsilon^2} + \frac{3}{2\varepsilon^4}}}.$$

Then, completing both squares,

$$\frac{|\langle w^*, y_i \rangle|}{\|w^*\| \, \|y_i\|} > \varepsilon \frac{1}{\left( 1 + \frac{1/4}{\varepsilon} \right) \left( \sqrt{2} + \frac{\sqrt{3/2}}{\varepsilon^2} \right)}$$

$$= \varepsilon^4 \frac{1}{(\varepsilon + 1/4) \left( \sqrt{2}\varepsilon^2 + \sqrt{3/2} \right)}.$$

As $\varepsilon < 1/2$ by Lemma B.7, we can bound the fraction below by some constant $c_1 \approx 0.845$. $\square$

Finally, we prove Theorem 1.2 restated below.

**Theorem.** *There exists a universal constant $c \geq 1$ such that for any fixed $\varepsilon > 0$, there exists an $n^{O((1/\varepsilon)^c)}d$ time algorithm that correctly clusters all $\varepsilon$-APS 2-means instances.*

*Proof.* The correctness of Algorithm B.3 for all $\varepsilon$-APS 2-means clustering instances in which each cluster has at least 4 points follows from Lemmas B.6, B.2, and B.9. On the other hand, the optimal 2-means clustering where one of the clusters has at most 3 points can be calculated in $O(n^4 d)$ time. An algorithm that returns the better of these two solutions then correctly clusters all $\varepsilon$-APS 2-means instances. $\qquad\square$

## Appendix C  $k$-means clustering for general $k$

We assume that our instance has balance parameter $\beta$. Our algorithm takes in as input the set of points $X$ and $k$, and outputs a clustering of all the points.

**Algorithm C.1.** ────────────────────────────────────────

**Input:** $X = \{x_1, \ldots, x_n\}$, $k$.
 1: **for all** pairs $a, b$ of distinct points in $\{x_i\}$ **do**
 2:     Let $r = \|a - b\|$ be our guess for $\rho$
 3:     **procedure** INITIALIZE
 4:         Create graph $G$ on vertex set $\{x_1, \ldots, x_n\}$ where $x_i$ and $x_j$ have an edge iff $\|x_i - x_j\| < r$
 5:         Let $a_1, \ldots, a_k \in \mathbb{R}^d$ where $a_i$ is the mean of the $i$th largest connected component of $G$
 6:     **procedure** ASSIGN
 7:         Let $C_1, \ldots, C_k$ be the clusters obtained by assigning each point in $X$ to the closest $a_i$
 8:     Calculate the $k$-means objective of $C_1, \ldots, C_k$
 9: Return clustering with smallest $k$-means objective found above

────────────────────────────────────────

The main theorem of this section states that the above poly-time algorithm exactly recovers optimal clusters $C_1, C_2, \ldots, C_k$ for instances with $\rho$ large enough in terms of $\varepsilon$, $\Delta$, and $\beta$.

**Theorem C.2.** *Algorithm C.1 recovers $C_1, \ldots, C_k$ for any $(\rho, \Delta, \varepsilon)$-separated instance with $\rho = \Omega\left(\frac{\Delta}{\varepsilon^2} + \frac{\beta\Delta}{\varepsilon}\right)$.*

The rest of the section is devoted to proving Theorem C.2. Define the following regions of $\mathbb{R}^d$ for every pair $i, j$. Given $i, j$, let $C_i, C_j$ be the corresponding clusters with means $\mu_i, \mu_j$. Let $u = \frac{\mu_i - \mu_j}{\|\mu_i - \mu_j\|}$ be the unit vector in the inter-mean direction and $p = \frac{\mu_i + \mu_j}{2}$ be the point between the two means. We first define formally $S_{i,j}^{(\mathrm{cone})}$ which was described in the introduction (the feasible region) and two other regions of the clusters that will be useful in the analysis (see Figure 1b). We observe that $C_i$ belongs to the intersection of all the cones $S_{i,j}^{(\mathrm{cone})}$ over $j \neq i$.

**Definition C.3.** Let $S_{i,j}^{(\mathrm{cone})} = \{x \in \mathbb{R}^d \mid \|(x - (\mu_i - \Delta u))_{(V)}\| < \frac{1}{\varepsilon}\langle x - (\mu_i - \Delta u), u\rangle\}$. The nice area of $i$ with respect to $j$ is defined as all points in the cap of $S_{i,j}^{(\mathrm{cone})}$ "above" $\mu_i$ i.e., $S_{i,j}^{(\mathrm{nice})} = \{x \in S_{i,j}^{(\mathrm{cone})} \mid \langle x - \mu_i, u\rangle \leq 0\}$. Finally, the good area of $C_i$ is the intersection of its nice areas i.e., $S_i^{(\mathrm{good})} = \bigcap_{j \neq i} S_{i,j}^{(\mathrm{nice})}$.

It suffices to prove the following two main lemmas. Lemma C.4 states that the ASSIGN subroutine correctly clusters all points given an initialization satisfying certain properties. Lemma C.5 states that the initialization returned by the INITIALIZE subroutine satisfies these properties when we guess $r = \rho$ correctly. As $\rho$ is only used as a threshold on edge lengths, testing the distances between all pairs of data points i.e. $\{\|a - b\| : a, b \in X\}$ suffices.

**Lemma C.4.** *For a $(\rho, \Delta, \varepsilon)$-separated instance with $\rho = \Omega(\Delta/\varepsilon^2)$, the ASSIGN subroutine recovers $C_1, C_2, \cdots C_k$ correctly when initialized with $k$ points $\{a_1, a_2, \ldots, a_k\}$ where $a_i \in S_i^{(\mathrm{good})}$.*

**Lemma C.5.** *For a $(\rho, \Delta, \varepsilon)$-separated instance with balance parameter $\beta$ and $\rho = \Omega(\beta\Delta/\varepsilon)$, the INITIALIZE subroutine outputs one point each from $\{S_i^{(\mathrm{good})} : i \in [k]\}$ when $r = \rho$.*

**Proof of Lemma C.4.** We will show that for any $a_i \in S_{i,j}^{(nice)}$, $a_j \in S_{j,i}^{(nice)}$, and $x \in C_i$, $x$ is closer to $a_i$ than to $a_j$. The following simple lemma describes the properties of the perpendicular bisector between $a_i$ and $a_j$. All these statements follow from the definitions of the nice regions and the angular separation (proof in Supplementary material).

**Lemma C.6.** *Suppose $\rho \geq \Delta/\varepsilon^2$. Then, for $a_i \in S_{i,j}^{(nice)}$ and $a_j \in S_{j,i}^{(nice)}$, we have*

(1) $\|(a_i - a_j)_{(u)}\| > \dfrac{\|(a_i - a_j)_{(V)}\|}{\varepsilon}$. (2) $\left\langle \dfrac{a_i + a_j}{2}, u \right\rangle \leq \dfrac{\Delta}{2}$. (3) $\left\| \left( \dfrac{a_i + a_j}{2} \right)_{(V)} \right\| \leq \Delta/\varepsilon$.

*Proof.*

1. We have $\|(a_i - a_j)_{(V)}\| \leq 2\Delta/\varepsilon$. On the other hand, $\rho \leq \|(a_i - a_j)_{(u)}\|$. Thus the inequality holds for $\rho > 2\Delta/\varepsilon^2$.

2. $\langle a_i + a_j, u \rangle = \langle a_i, u \rangle + \langle a_j, u \rangle \leq D_{i,j}/2 + (-D_{i,j}/2 + \Delta) = \Delta$. Multiplying by $1/2$ gives the desired inequality.

3. $\|(a_i + a_j)_{(V)}\| \leq \|(a_i)_{(V)}\| + \|(a_j)_{(V)}\| < 2\Delta/\varepsilon$. Multiplying by $1/2$ gives the desired inequality.

$\square$

We have all the properties we require from points in the nice region and we prove Lemma C.4 next. To do so we first rewrite the condition $\|x - a_i\| < \|x - a_j\|$ as $\frac{1}{2}\langle x - (a_i + a_j), a_i - a_j \rangle > 0$. Then we write each vector in terms of their projection on $u$ and $V$ and use Lemma C.6 to bound each of the terms.

*Proof of Lemma C.4.* It suffices to show that for any $a_i \in S_{i,j}^{(nice)}$, $a_j \in S_{j,i}^{(nice)}$, and $x \in C_i$, $\|x - a_i\| < \|x - a_j\|$. Then by Lemma C.6 above,

$$
\left\langle x - \frac{a_i + a_j}{2}, a_i - a_j \right\rangle = \left\langle x_{(u)}, (a_i - a_j)_{(u)} \right\rangle + \left\langle x_{(V)}, (a_i - a_j)_{(V)} \right\rangle
$$
$$
- \frac{1}{2} \left\langle ((a_i + a_j))_{(u)}, (a_i - a_j)_{(u)} \right\rangle - \frac{1}{2} \left\langle ((a_i + a_j))_{(V)}, (a_i - a_j)_{(V)} \right\rangle
$$
$$
> \|x_{(u)}\| \|(a_i - a_j)_{(u)}\| - \frac{1}{\varepsilon} \left( \|x_{(u)}\| - \frac{\rho}{2} \right) \varepsilon \|(a_i - a_j)_{(u)}\|
$$
$$
- \frac{\Delta}{2} \|(a_i - a_j)_{(u)}\| - \frac{\Delta}{\varepsilon} \varepsilon \|(a_i - a_j)_{(u)}\|
$$
$$
= \frac{\rho}{2} - \frac{3}{2}\Delta > 0
$$

where the inequality follows because of equality on the first term and Cauchy-Schwarz on the rest. So, for all $a_i \in S_{i,j}^{(nice)}$, $a_j \in S_{j,i}^{(nice)}$, and $x \in C_i$, $x$ is closer to $a_i$ than $a_j$. $\square$

**Proof of Lemma C.5.** We now show that `INITIALIZE` finds one point in each of the $k$ good areas. We first start by defining the "core region" of each cluster.

**Definition C.7** ($S^{(\text{core})}$). Let $S_i^{(\text{core})} = \{\, x \in \mathbb{R}^d \mid \|x - \mu_i\| < \Delta/\varepsilon \,\}$.

The core region is defined in such a way that most (at least $\beta/(1 + \beta)$ fraction) of the points in $C_i$ will belong to the connected component containing $S_i^{(\text{core})}$. Hence, any large connected component (in particular, the $k$ largest ones) must contain the core of one of the clusters. Further, since this component accounts for most points in $C_i$, the angular separation ensures that the empirical mean of the connected component is in $S_i^{(\text{good})}$. Before we move on to the proof, we state some properties of the connected components.

**Lemma C.8.**

1. *Any connected component only contains points from a single cluster.*

2. For all $i, j$, $S_i^{(core)} \supseteq S_{i,j}^{(nice)}$. In particular, $S_i^{(core)} \cap X$ is nonempty.

3. Fix $C_i$, $C_j$. Let $A_{i,j} = \{ x \in C_i \mid \langle x - \mu_i, \frac{\mu_i - \mu_j}{\|\mu_i - \mu_j\|} \rangle \leq \beta\Delta \}$. Then, $|A_{i,j}| \geq \frac{\beta}{1+\beta}|C_i|$.

4. For all $i, j$, $A_{i,j}$ is connected in $G$.

5. For all $i$, the largest component $K_i$ in cluster $C_i$ contains $A_{i,j}$ for all $j \neq i$.

*Proof.*

1. Let $x \in C_i$ and $y \in C_j$. Then $\|x - y\| \geq |\langle x - y, u \rangle| > \rho$, thus no edge connecting points in different clusters is added to $G$.

2. For $x \in S_{i,j}^{(nice)}$, $\|(x - \mu_i)_{(V)}\| < \frac{1}{\varepsilon}(\Delta - \|(x - \mu_i)_{(u)}\|)$, hence $\|x - \mu_i\| < \Delta/\varepsilon$. An argument similar to that in Lemma B.6 shows that $S_{i,j}^{(nice)} \cap X$ is nonempty and hence $S_{i,j}^{(core)} \cap X$ is nonempty.

3. $\mu_i$ is the mean of the points in cluster $C_i$. By an averaging argument, $|A_{i,j}|\Delta - (|C_i| - |A_{i,j}|)\beta\Delta \geq 0$. Rearranging, $|A_{i,j}| \geq \frac{\beta}{1+\beta}|C_i|$.

4. We show that all of $A_{i,j}$ is connected. Recall $S_{i,j}^{(nice)}$ is nonempty; let $x \in S_{i,j}^{(nice)}$. For $y \in A_{i,j}$, $\|x - y\|^2 = \|(x - y)_{(u)}\|^2 + \|(x - y)_{(V)}\|^2 < ((\beta + 1)\Delta)^2 + ((\beta + 1)\Delta/\varepsilon)^2$. Thus for $\rho = \Omega(\beta\Delta/\varepsilon)$, all of $A_{i,j}$ is connected.

5. By part 2, $S_i^{(core)} \cap I$ is nonempty; let $a_i \in S_i^{(core)} \cap I$ and let $K_i$ be the connected component containing $a$. As $S_i^{(core)} \supseteq S_{i,j}^{(nice)}$, and $S_{i,j}^{(nice)} \cap I$ is nonempty, $K_i$ contains $A_{i,j}$ by the above argument. The size of this connected component is at least $\beta/(1 + \beta)|C_i|$. As $\beta \geq 1$, $K_i$ is the largest component in the cluster.

$\square$

Lemma C.9 states that each of the $k$ largest component must lie in a different cluster which proves that each of the $k$ means $\{ a_1, \ldots, a_k \}$ come from a different cluster. While Lemma C.10 states that each $a_i$ lie inside a good region. Together, they imply Lemma C.5, i.e. each $a_i$ comes from a different good area.

**Lemma C.9.** *The set of $k$ largest components of $G$ contains the largest component of each cluster.*

*Proof.* Let $K_i$ be the largest component in $C_i$ and let $K_j'$ be a component in $C_j$ that is not the largest. Then by the $\beta$ parameter, $|K_i| \geq \frac{\beta}{1+\beta}|C_i| > \frac{1}{1+\beta}|C_j| \geq |K_j'|$. It follows that the $k$ largest connected components are $K_1, K_2, \ldots, K_k$. $\square$

**Lemma C.10.** *The mean of points in $K_i$ lies in $S_i^{(good)}$.*

*Proof.* Fix $i$ and consider $K_i$. For $j \neq i$, $K_i \subseteq A_{i,j}$. Additionally, $K_i \subseteq S_{i,j}^{(cone)}$ a convex set. It follows that the mean of points in $K_i$ lies in $S_{i,j}^{(cone)}$. As $K_i \supseteq S_i^{(core)} \supseteq S_{i,j}^{(nice)}$, the points not contained in $K_i$ have $\langle x - \mu, u \rangle > 0$ and hence the mean of points in $K_i$ lies in $S_{i,j}^{(nice)}$. As this holds for each $j \neq i$, the mean of points in $K_i$ lies in $S_i^{(good)}$. $\square$

# Appendix D  Robust $k$-means

We show in this section that a simple extension of algorithm 3.1 does well even in the presence of adversarial noise for instances with $(\rho, \Delta, \varepsilon)$-separation for large enough $\rho$. Specifically, we consider the following model.

Let $X = \{x_1, \ldots, x_n\} \subset \mathbb{R}^d$ be a $k$-means clustering instance with optimal clustering $C_1, \ldots, C_k$. We call $X$ the set of *pure* points. An additional set of at most $\eta n$-many *impure* points $Z \subset \mathbb{R}^d$ is added by an adversary. Our goal is to find a clustering of $X \cup Z$ that agrees with $C_1, \ldots, C_k$ on the pure points.

Let $w_{\max} = \max|C_i|/n$ and let $w_{\min} = \min|C_i|/n$ be the maximum and minimum weight of clusters. We will assume that $\eta < w_{\min}$.

**Algorithm D.1.** ───────────────────────────────────────

**Input:** $X \cup Z, r, t$

 1: **procedure** INITIALIZE
 2:      Create graph $G$ on $I \cup Z$ where points are connected by an edge iff they are at distance $\leq r$
 3:      Remove vertices with vertex degree $\leq t$
 4:      Let $a_1, \ldots, a_k \in \mathbb{R}^d$ where $a_i$ is the mean of the $i$th largest connected component of $G$
 5: **procedure** ASSIGN
 6:      Let $C_1, \ldots, C_k$ be the clusters obtained by assigning each point in $I \cup Z$ to the closest $a_i$

───────────────────────────────────────────────────────────

**Theorem D.2.** *Given $X \cup Z$ where $X$ satisfies $(\rho, \Delta, \varepsilon)$-separation for*

$$\rho = \Omega\left(\frac{\Delta}{\varepsilon^2}\left(\frac{w_{\max} + \eta}{w_{\min} - \eta}\right)\right)$$

*and $\eta < w_{\min}$, there exists values of $r, t$ such that Algorithm D.1 returns a clustering consistent with $C_1, \ldots, C_k$ on $X$. Algorithm D.1 runs in time $O(n^2 d)$.*

Just as in Theorem 3.2, this algorithm uses $r$ and $t$ as thresholds. Hence, it is possible to guess $r$ from the $\binom{n}{2}$ pairwise edge lengths and $t$ from $[n]$.

Fix the following parameters.

$$\alpha = 2\left(\frac{w_{\max} + \eta}{w_{\min} - \eta}\right), \qquad r = (\alpha + 1)\Delta(1 + 2/\varepsilon), \qquad t = w_{\min}\eta\frac{\alpha}{\alpha + 1}.$$

**Definition D.3.** Define the following regions.

- For $i \neq j$, let $S_{i,j}^{(\text{e nice})} = \{\, x \in S_{i,j}^{(\text{cone})} \mid \langle x - \mu_i, u \rangle \leq \alpha \,\}$.

- For $i \neq j$, let $S_{i,j}^{(\text{r e nice})} = \{\, x \in \mathbb{R}^d \mid d(x, S_{i,j}^{(\text{e nice})}) \leq r \,\}$ be the robust nice region.

- Let $S_i^{(\text{r good})} = \bigcap_{j \neq i} S_{i,j}^{(\text{r e nice})}$ be the robust good region.

It suffices to prove the following two lemmas.

**Lemma D.4.** *Given $X \cup Z$ where $X$ satisfies the $(\rho, \Delta, \varepsilon)$-condition for $\rho = \Omega(\alpha\Delta/\varepsilon^2)$ and $\eta < w_{\min}$, the ASSIGN subroutine recovers $C_1, \ldots, C_k$ correctly when initialized with $k$ points $\{\, a_1, \ldots, a_k \,\}$ where $a_i \in S_i^{(\text{r good})}$.*

**Lemma D.5.** *Given $X \cup Z$ where $X$ satisfies the $(\rho, \Delta, \varepsilon)$-condition for $\rho = \Omega(\alpha\Delta/\varepsilon^2)$ and $\eta < w_{\min}$, for the choices of $r$ and $t$ as above, the INITIALIZE subroutine provides $k$ points $\{\, a_1, \ldots, a_k \,\}$ where $a_i \in S_i^{(\text{r good})}$.*

## D.1   Proof of Lemma D.4

We will show that for any $a_i \in S_i^{(\text{r good})}$, $a_j \in S_j^{(\text{r good})}$ and $x \in C_i$, $x$ is closer to $a_i$ than $a_j$.

The following lemma describes the properties of the perpendicular bisector between $a_i$ and $a_j$.

**Lemma D.6.** *There is some $\rho \geq \Omega(\alpha\Delta/\varepsilon^2)$ such that for $a_i \in S_{i,j}^{(\text{r e nice})}$ and $a_j \in S_{j,i}^{(\text{r e nice})}$,*

1. $\varepsilon\|(a_i - a_j)_{(u)}\| \geq \|(a_i - a_j)_{(V)}\|$,

2. $\langle\frac{a_i + a_j}{2}, u\rangle < (\alpha + 1)\Delta/2 + r$,

3. $\left\| \left( \frac{a_i + a_j}{2} \right)_{(V)} \right\| \le (\alpha + 1)\Delta/\varepsilon + r.$

*Proof.*

1. By triangle inequality, $\|(a_i - a_j)_{(V)}\| \le 2((\alpha+1)\Delta/\varepsilon + r)$. On the other hand, $\|(a_i - a_j)_{(u)}\| \ge \rho - 2r$. Thus the inequality holds for $\rho \ge 2r + \frac{2}{\varepsilon}((\alpha+1)\Delta/\varepsilon + r)$.

2. $\langle a_i + a_j, u \rangle = \langle a_i, u \rangle + \langle a_j, u \rangle < (D_{i,j}/2 + \alpha\Delta + r) + (-D_{i,j}/2 + \Delta + r) = (\alpha+1)\Delta + 2r$. Multiplying by $1/2$ gives the desired inequality.

3. $\|(a_i + a_j)_{(V)}\| \le \|a_{i,(V)}\| + \|a_{j,(V)}\| < 2((\alpha+1)\Delta/\varepsilon + r)$. Multiplying by $1/2$ gives the desired inequality.

$\square$

We now prove Lemma D.4.

*Proof of Lemma D.4.* It suffices to show that for any $a_i \in S_i^{(\text{r good})}$, $a_j \in S_j^{(\text{r good})}$ and $x \in C_i$, $\|x - a_i\| < \|x - a_j\|$. By definition $S_i^{(\text{r good})} \subseteq S_{i,j}^{(\text{r e nice})}$ for all $j$. Thus by Lemma D.6 above,

$$
\left\langle x - \frac{a_i + a_j}{2}, a_i - a_j \right\rangle = \left\langle x_{(u)}, (a_i - a_j)_{(u)} \right\rangle + \left\langle x_{(V)}, (a_i - a_j)_{(V)} \right\rangle
$$

$$
- \left\langle \left( \frac{a_i + a_j}{2} \right)_{(u)}, (a_i - a_j)_{(u)} \right\rangle - \left\langle \left( \frac{a_i + a_j}{2} \right)_{(V)}, (a_i - a_j)_{(V)} \right\rangle
$$

$$
\ge \|x_{(u)}\|\|(a_i - a_j)_{(u)}\| - \frac{1}{\varepsilon}\left( \|x_{(u)}\| - \rho/2 \right)\varepsilon\|(a_i - a_j)_{(u)}\|
$$

$$
- ((\alpha+1)\Delta/2 + r)\|(a_i - a_j)_{(u)}\| - ((\alpha+1)\Delta/\varepsilon + r)\varepsilon\|(a_i - a_j)_{(u)}\|
$$

$$
= \rho/2 - \left( \frac{3}{2}(\alpha+1)\Delta + (1+\varepsilon)2r \right).
$$

Thus for some $\rho \ge \Omega(\alpha\Delta/\varepsilon^2)$, this quantity is greater than zero, i.e., for all $a_i \in S_i^{(\text{r good})}$, $a_j \in S_j^{(\text{r good})}$, and $x \in C_i$, $x$ is closer to $a_i$ than to $a_j$. $\square$

## D.2 Proof of Lemma D.5

We prove Lemma D.5 using the below proof outline.

*Proof outline.*

1. For any $i \ne j$, the set of vertices $S_{i,j}^{(\text{e nice})} \cap X$ forms a clique and the size of this clique is greater than $t$. In particular, no vertex in $S_{i,j}^{(\text{e nice})}$ is deleted.

2. Fix $i$. For all $j \ne i$, the vertices $S_{i,j}^{(\text{e nice})} \cap X$ belong to a single connected component. Let $K_i$ be this connected component.

3. Before vertex deletion (and after), no vertex is adjacent to pure points from different clusters.

4. After vertex deletion, every remaining point lies in $S_i^{(\text{r good})}$ for some $i$. Hence by part 2, every connected component contains pure points from at most a single cluster. In particular, $K_1, \ldots, K_k$ are distinct.

5. Consider $K_i$ as above and an arbitrary connected component $K$ not in $K_1, \ldots, K_k$. Then $|K_i| > |K|$. In particular, the $k$ largest components of $G$ are $K_1, \ldots, K_k$.

6. The mean of $K_i$ lies in $S_i^{(\text{r good})}$.

$\square$

*Proof.*

1. The diameter of $S_{i,j}^{(\text{e nice})}$ is $\text{diam}(S_{i,j}^{(\text{e nice})}) \leq (\alpha+1)\Delta(1+2/\varepsilon) = r$. Thus every pair of points in this region is connected. Recall that $\mu_i$ is the mean of the pure points in cluster $C_i$. By an averaging argument, $|S_{i,j}^{(\text{e nice})} \cap X|\Delta - (|C_i| - |S_{i,j}^{(\text{e nice})} \cap X|)\alpha\Delta \geq 0$. Rearranging, $|S_{i,j}^{(\text{e nice})} \cap I| \geq \frac{\alpha}{\alpha+1}|C_i| \geq \frac{\alpha}{\alpha+1}nw_{\min} = t$.

2. Fix $j$. Let $j \neq i$. Recall $S_{i,j}^{(\text{nice})} \cap X$ is nonempty; let $x \in S_{i,j}^{(\text{nice})} \cap X$. Then $\|x - \mu_i\| \leq \Delta/\varepsilon$. We show that for any $j' \neq i$, the connected component containing $x$ contains $S_{i,j'}^{(\text{e nice})} \cap X$. Let $y \in S_{i,j'}^{(\text{e nice})} \cap X$. Then $\|y - x\| \leq \|y - \mu_i\| + \|x - \mu_i\| \leq (\alpha+1)\Delta/\varepsilon + \alpha\Delta + \Delta/\varepsilon \leq \Delta(\alpha+1)(1+2/\varepsilon) = r$.

3. Pure points in different clusters are at distance at least $\rho$ whereas two vertices sharing a neighbor must be at distance at most $2r$. Thus the inequality holds for $\rho \geq \Omega(\alpha\Delta/\varepsilon)$.

4. Suppose a point $p$ not in $\bigcup_i S_i^{(\text{r good})}$ remains after vertex deletion. By part 3 above, $p$ can only be connected to pure points in a single cluster. Suppose it is connected to pure points in cluster $C_i$. By assumption, there exists a $j$ such that $p \notin S_{i,j}^{(\text{r e nice})}$. We bound the degree of $p$ above by the number of points in $X \setminus S_{i,j}^{(\text{e nice})}$ and the $\eta n$-many impure points, i.e., $\deg(x) \leq \eta n + \frac{|C_i|}{\alpha+1} \leq n(\eta + \frac{w_{\max}}{\alpha+1})$. By our choice of $t$, we have that $\deg(p) < t$. Thus $p$ is deleted and all remaining points lie in $\bigcup_i S_i^{(\text{r good})}$.

   For any $i, j$, the minimum distance between $S_i^{(\text{r good})}$ and $S_j^{(\text{r good})}$ is at least $\rho - 2r$. For some $\rho \geq \Omega(\alpha\Delta/\varepsilon)$ then, the distance between these regions is greater than $\rho - 2r > r$ and no connected component contains pure points from multiple clusters.

5. As in part 2, the size of $K_i$ is bounded below by the averaging argument $|K_i| \geq \frac{\alpha}{\alpha+1}|C_i|$. Let $K$ belong to cluster $C_j$. Then by part 5, the size of the connected component $K$ is bounded above by the number of remaining points after $K_j$ is removed and the $\eta n$-many impure points, i.e., $|C_j| \leq \frac{1}{\alpha+1}|C_j| + \eta n$. Then by our choice of $\alpha$, $|K| < |K_i|$.

6. By part 4, $K_i \subseteq S_i^{(\text{r good})}$. As $S_i^{(\text{r good})}$ is convex, the mean of $K_i$ also lies in $S_i^{(\text{r good})}$.

$\square$

## Appendix E   Experimental Results

Table 3: Values of $\varepsilon$ satisfying Lemma 2.4

| Dataset | Minimum $\varepsilon$ | Average $\varepsilon$ | Maximum $\varepsilon$ |
|---|---|---|---|
| Wine | 0.0115 | 0.0731 | 0.191 |
| Wine (normalized) | 0.000119 | 0.0394 | 0.107 |
| Iris | 0.00638 | 0.103 | 0.256 |
| Iris (normalized) | 0.00563 | 0.126 | 0.237 |
| Banknote Auth. | 0.00127 | 0.00127 | 0.00127 |
| Banknote (norm.) | 0.00175 | 0.00175 | 0.00175 |
| Letter Recognition | 3.22e-05 | 0.0593 | 0.239 |
| Letter Rec. (norm.) | 8.49e-06 | 0.0564 | 0.247 |

**Ground truth recovery.** The clustering returned by our algorithm recovers well ($\approx 97\%$) the solution returned by Lloyd's algorithm initialized with the ground truth for Wine, Iris, and Banknote Authentication across normalized and unnormalized datasets.

Table 4: Values of $(\rho, \varepsilon, \Delta)$ satisfied by $(1 - \eta)$-fraction of points

| Dataset | $\eta$ | $\varepsilon$ | minimum $\rho/\Delta$ | average $\rho/\Delta$ | maximum $\rho/\Delta$ |
|---|---|---|---|---|---|
| Wine | 0.05 | 0.1 | 0.355 | 0.992 | 2.19 |
| | | 0.01 | 0.374 | 1 | 2.2 |
| | 0.1 | 0.1 | 0.566 | 1.5 | 3.05 |
| | | 0.01 | 0.609 | 1.53 | 3.07 |
| Wine (normalized) | 0.05 | 0.1 | | | |
| | | 0.01 | 0.399 | 1.06 | 2.29 |
| | 0.1 | 0.1 | 0.451 | 1.3 | 2.66 |
| | | 0.01 | 0.735 | 1.96 | 3.62 |
| Iris | 0.05 | 0.1 | 0.156 | 2.47 | 5.37 |
| | | 0.01 | 0.263 | 2.88 | 6.43 |
| | 0.1 | 0.1 | 0.398 | 4.35 | 7.7 |
| | | 0.01 | 0.496 | 5.04 | 9.06 |
| Iris (normalized) | 0.05 | 0.1 | 0.0918 | 1.89 | 3.08 |
| | | 0.01 | 0.213 | 2.21 | 3.4 |
| | 0.1 | 0.1 | 0.223 | 3.74 | 7.12 |
| | | 0.01 | 0.391 | 4.42 | 8.3 |
| Banknote Auth. | 0.05 | 0.1 | 0.0731 | 0.0731 | 0.0731 |
| | | 0.01 | 0.198 | 0.198 | 0.198 |
| | 0.1 | 0.1 | 0.264 | 0.264 | 0.264 |
| | | 0.01 | 0.398 | 0.398 | 0.398 |
| Banknote (norm.) | 0.05 | 0.1 | | | |
| | | 0.01 | 0.197 | 0.197 | 0.197 |
| | 0.1 | 0.1 | 0.246 | 0.246 | 0.246 |
| | | 0.01 | 0.474 | 0.474 | 0.474 |
| Letter Recognition | 0.05 | 0.1 | | | |
| | | 0.01 | 0.168 | 2.06 | 6.96 |
| | 0.1 | 0.1 | 0.018 | 2.19 | 7.11 |
| | | 0.01 | 0.378 | 3.07 | 11.4 |
| Letter Rec. (norm.) | 0.05 | 0.1 | | | |
| | | 0.01 | 0.157 | 1.97 | 7.14 |
| | 0.1 | 0.1 | | | |
| | | 0.01 | 0.378 | 2.92 | 11.2 |