[Reviews · NeurIPS 2017]

Reviewer 1



The authors give provably efficient algorithms to recover optimal k-means clusters on input instances that are stable with respect to additive perturbations. This notion is incomparable to scale invariant multiplicative perturbations that were studied earlier [5, 12]. The main technical contribution of this paper is to remove the diam(X) dependency in the exponent from the running time of previous algorithms by Ben David and others for additive perturbation stable instances. As a bonus, this also makes their result robust to a constant fraction of outliers. Overall, the paper is well-written and easy to follow. In the experimental section, the authors show that their algorithm gives a better initialization than k-means++. I guess this is because their algorithm is robust to outliers whereas k-means++ is extremely sensitive to outliers. However, strangely the last 2 columns of Table 1 suggest that when this initialization is followed by Lloyd's iterations, then k-means++ gets comparable (or sometimes better).

Reviewer 2



The authors consider stability of the Euclidean k-means problem, which is well used method in various data analysis. As the measure for the stability, they consider epsilon-APS. I have questions for your methods: - The various clusterability measures have been proposed as surveyed by M. Ackerman et al (M. Ackerman et al., PMLR. 2009). Comparing these measures, what is an advantage of the measure you used? - Can't we use other measures as an initialization method of Lloyd’s method? If we can, how about the performance of them comparing to epsilon-APS. - The stability of clustering result depends on the parameters rho, delta, and epsilon. It is unclear for me how to find a set of appropriate parameters before we perform clustering. - I guess that there are datasets that cannot satisfy epsilon-APS. Can I know the existence of epsilon-APS condition in the dataset before clustering?

Reviewer 3



The authors propose a notion of additive perturbation stability (APS) for Euclidean distances that maintain the optimal k-means clustering solution when each point in the data is moved by a sufficiently small Euclidean distance. I think the paper is rather interesting; however, the results of the paper are not very surprising. Here are my comments regarding the paper: (1) To my understanding, the results of Theorem 1.2 are only under the condition of APS. They only hold for the case of k=2 components and may lead to exponential dependence on $k$ components for large $k$. However, under the additional margin condition between any two pairs of cluster, we will able to guarantee the existence of polynomial algorithm on $k$. Can you provide a high level idea of how this additional assumption actually helps? Is it possible to have situation without that margin condition that there exists no algorithm that is polynomial in terms of $k$? (2) I find the term $\Delta = (1/2-\epsilon)D$, where $D$ is the maximum distance between any pairs of means, is interesting. It also has a nice geometric meaning, which is the distance between any center to the apex of cone. I wonder whether this term $\Delta$ is intrinsic to the setting of APS? May it still be available under other perturbation notions? (3) In the result of Theorem 1.3 and later Theorem 3.2, we both require the condition that $\rho$, which is a threshold for the margin between any two pairs of cluster, to be at least the oder of $\Delta/\epsilon^{2}$. As $\epsilon$ is sufficiently small, which is also the challenging theme of APS, the term $\Delta/\epsilon^{2}$ becomes very big, i.e., the distances between any two pairs of cluster become very large. The fact that there is an algorithm running in time polynomial of $k$ components, sample size $n$, and dimension $d$ under that setting is not surprising. Is it possible to improve the lower bound of the margin between any two pairs of cluster? (4) In all the results with running time of the paper, the authors only provide the bound in terms of $n$, $d$, and $k$. I wonder about the constants that are along with these terms. How do they change with margin $\rho$ and $\epsilon$-APS? (5) As the authors indicate in the paper, data in practice may have very small value of $\epsilon$. It seems that the conditions of the results with polynomial algorithms in the paper, e.g. Theorem 3.2, will not be satisfied by real data. (6) It seems to me that the results in the paper can be extended to other distances that are different from Euclidean distance. The geometric explanation of additive perturbation stability may be different, which also leads to the modifications of other notions and results in the paper. Such extensions will be very useful in case that we want to work with kernel K-means to deal with sophisticated structures of data.